# Effectiveness and Cost-Effectiveness of Mental Health Interventions Delivered by Frontline Health Care Workers in Emergency Health Services: A Systematic Review and Meta-Analysis

**DOI:** 10.3390/ijerph192315847

**Published:** 2022-11-28

**Authors:** Min Peng, Tao Xiao, Ben Carter, Pan Chen, James Shearer

**Affiliations:** 1King’s Health Economics, Institute of Psychiatry Psychology and Neuroscience, King’s College London, London SE5 8AF, UK; 2Department of Emergency Response and Preparedness, The Second Xiangya Hospital of Central South University, Changsha 410011, China; 3Psychological Rescue Branch, China Association for Disaster and Emergency Rescue Medicine, Haidian District, Beijing 100080, China; 4Department of Biostatistics and Health informatics, Institute of Psychiatry Psychology and Neuroscience, King’s College London, London SE5 8AF, UK; 5Hunan Cancer Hospital, The Affiliated Cancer Hospital of Xiangya School of Medicine, Central South University, Changsha 410011, China

**Keywords:** mental health intervention, natural disasters, frontline health care workers, effectiveness, cost effectiveness

## Abstract

This systematic review is to evaluate the effectiveness and cost-effectiveness of mental health interventions delivered by frontline health care workers in disasters and public health emergencies. Six databases and trial registries were searched, and manual searches were conducted. Of the 221 studies identified, 21 were included. Meta-analyses assessed differences between the intervention and control in terms of PTSD outcomes. Eleven studies of 1802 participants were incorporated in the meta-analysis. Interventions delivered or prompted by specialist health care workers showed significant and large effects in improving PTSD-related symptoms with a SMD = 0.99 (95% CI: 0.42–1.57, *p* = 0.0007). Interventions delivered or prompted by frontline non-specialist health care workers showed significant but small effects in improving PTSD-related symptoms with SMD of 0.25 (95% CI: 0.11–0.39; *p* = 0.0007). The results showed that most mental health interventions delivered by frontline health care workers effectively supported affected people. Mental health interventions delivered by mental health care professionals are effective in reducing PTSD-related disorders in natural disasters. Future adequately powered RCTs are needed to evaluate the effectiveness of mental health interventions delivered by trained non-specialists. Economic modelling may be useful to estimate cost effectiveness in low- and middle-income countries given the difficulties of conducting studies in disaster and emergency settings.

## 1. Introduction

Natural disasters have killed approximately 60,000 people every year, on average, during the last two decades [1]. In 2021, natural disasters caused approximately 10,492 deaths and affected 101.8 million people globally, with an estimated global economic loss of $343 billion USD [2]. The World Health Organization has identified disaster-related mental health issues as among the most pressing public health issues [3]. Natural disasters (such as floods, bushfires, and earthquakes) and public emergencies (such as pandemics, wars, and civil disorders) have negative mental health implications for those who are directly or indirectly affected [4]. Due to exposure to traumatic events and chronic stress, various mental health problems, such as anxiety, depression, post-traumatic stress disorder (PTSD), and bipolar emotional disorders, occur during disasters. PTSD is the most common mental health problem following a disaster [5]. According to the American Psychiatric Association [6], when an individual lives through or witness traumatic incidents or is being threatened by death, experiencing severe injury, or sexual abuse, post-traumatic stress disorder (PTSD) may occur. Directly experiencing the traumatic incident, seeing it happen to others, hearing that it happened to a family member or close acquaintance, or indirectly being exposed due to work-related responsibilities are all forms of exposure. Examples of traumatic situations include war, physical assault, terrorist strikes, and natural catastrophes. PTSD symptoms include (a) reliving the trauma in upsetting memories, flashbacks, or frequent dreams or nightmares; (b) avoiding situations or activities that bring up the traumatic event; (c) diminished responsiveness (emotional anaesthesia or numbing); and (d) feelings of detachment and estrangement from others. Chronic physiological arousal can also cause symptoms such as an exaggerated startle (see survivor guilt).

The prevalence of mental health disorders in disaster-affected areas has been reported to be double to triple that in the general population [7]. According to a systematic review published in 2019, the prevalence of severe mental disorders in conflict settings (schizophrenia, bipolar disorder, depression, anxiety, and post-traumatic stress disorder) was 22.1%, and one out of every 11 people (9%) exposed to conflict in the previous ten years would develop a moderate or severe mental disorder [8]. A review carried out between January 2009 and March 2013 from 90 refugee camps across 15 low- and middle-income countries revealed that 77% of healthcare visits were for mental substance use disorders, psychotic disorders, and moderate and severe depression, anxiety, and PTSD [9].

In emergency situations, post-traumatic stress disorder (PTSD) is the most common mental health issue. The prevalence of PTSD among direct victims is estimated to be 30–40%, while it is 10–20% among rescue professionals and 5–10% in the general public [10]. Mental health issues can become chronic if not treated. According to research conducted in a refugee camp on the Turkish–Syrian border, 24–30% of those who were exposed to the Syrian Civil War had PTSD and depression co-morbidly, 4–6% solely had PTSD, and 19% only had depression three years later [11].

Early mental health care may reduce negative mental health effects for those affected by emergencies [12,13]. Interventions suitable for use in disaster and public emergency situations need to be brief and culturally relevant, with proven effectiveness and cost-effectiveness [14]. Frontline healthcare workers must be able to provide rapid mental health support in addition to physical healthcare to affected people as first responders in emergencies [15].

Previous systematic reviews have been unable to offer clear conclusions to guide implementation or health policy on this topic because they needed to focus on all these disorders together, as opposed to one or some only, for several reasons: (1) they targeted certain clinical populations such as people suffering from depression or anxiety disorders, particularly post-traumatic stress disorder (PTSD) [16,17,18]; (2) they focused on the effects of specific psychiatric therapies such as cognitive behavioral therapy, eye movement desensitization, and reprocessing (EMDR) or therapeutic components [18,19,20,21,22]; and (3) studies did not include evidence from low- and middle-income countries (LAMICs) or underrepresented LAMICS [23,24].

The current understanding of which psychological interventions and care models can improve mental wellbeing in disasters/emergencies is incomplete, based on insufficiently robust and clear evidence [25]. This study aimed to assess the effectiveness and cost-effectiveness of all types of mental health interventions delivered by frontline healthcare workers during natural disasters and in public emergency settings.

## 2. Methods

This systematic review and associated meta-analysis were implemented according to the Cochrane guidelines on systematic reviews of observational studies and according to PRISMA-P. The PRISMA-P checklist is illustrated in Figure 1. The protocol was registered in the International Prospective Register of Systematic Reviews (CRD 42021266648).

### 2.1. Eligibility Criteria

Studies focusing on mental health interventions involving preparedness, response, and rehabilitation in natural disaster and emergency settings were eligible, except for qualitative studies, uncontrolled studies, conference presentations, studies of personal traumatic events, and medical emergency studies. Mental health interventions refer to general approaches to promoting awareness of mental health or measures tailored to deal with a specific mental health disorder, such as depression or anxiety. The types of interventions ranged from counselling to psychotherapy, psychoeducation, behavioral activation, and cognitive behavioral therapy (CBT). No restrictions were placed on the method used to assess mental health, with data ranging from self-report surveys to objective assessments recorded using established methodologies.

### 2.2. Search Strategy

Six databases (Embase, PsycINFO, Epub, Global Health, APA PsycInfo, and MEDLINE) were searched from their inception to December 2021. Searches were conducted by combining combinations of the five categories of keywords (Appendix A). In addition, Google Scholar and WHO databases were manually searched, and 31 additional studies were identified. There were no data limitations in this study.

### 2.3. Selection Process and Data Extraction

The search results were exported to EndNote 20, where duplicates were removed. The included studies were screened manually to identify other potentially relevant studies. The characteristics of the studies included are summarized in Table 1. The primary investigator (MP) reviewed the search strategy. Following the above inclusion and exclusion criteria, two reviewers (MP and JS) independently reviewed relevant studies for potential inclusion. Any differences in opinions were resolved through discussion until a consensus was reached. This process minimized the risk of bias in the decision to include or exclude studies. A PRISMA flowchart is shown in Figure 1.
ijerph-19-15847-t001_Table 1Table 1Summary characteristics of included studies.Author and YearCountry of OriginDisaster and YearStudy Design and ParticipantsInterventionComparatorsPrimary Outcome Measures1. (Başoğlu, 2005) [14]Turkey1999 EarthquakeStudy design: Randomised control trial.N = 59. Survivors of earthquakeMean age: 36.3 years (16–65 years)Experimental Group/s: intervention *n* = 31; waitlist control *n* = 28Brief behavioral treatmentWaitlist controlClinician-Administered PTSD Scale (CAPS; Blake et al., 1995) [26]2. (Wolmer, 2005) [27]Turkey1999 EarthquakeStudy design: Controlled before and after study.N = 287. Students studied in three schools located in the disaster area.Mean age: 11.5 years (Children aged 9–17 years)Experimental Group/s: intervention *n* = 67; control *n* = 220.School reactivation programNo treatmentThe Child Post-Traumatic Stress Disorder Reaction Index (CPTSD-RI; Pynoos et al., 1987) [28]3. (Steinmetz, 2012) [29]USAHurricane Study design: Randomised control trial.N = 56. Survivors from previous stress study.Mean age: 43 yearsExperimental Group/s: intervention *n* = 18; control (UC) *n* = 19; information only *n* = 19Internet-based psychoeducationUsual care or Information onlyTrauma Screening Questionnaire (Brewin et al., 2002) [30]4. (Zang, 2013) [31]ChinaEarthquakeStudy design: Randomised control trial.N = 22. People severely affected by earthquakeMean age: 55.73 yearsExperimental Group/s: intervention *n* = 11; waitlist control *n* = 11Narrative Exposure Therapy (NET)Waitlist controlImpact of Event Scale-Revised (IES-R) (Weiss, 2007) [32]; General Health Questionnaire-28 (Goldberg and Hillier, 1979) [33]5. (Adams, 2013) [34]USALarge-scale community disasterStudy design: Controlled before and after studyN = 215. Primary care paediatricians.Mean age: n/aExperimental Group/s: intervention *n* = 137; control *n* = 78RCI trainingN/APractice change survey6. (Zang, 2014) [35]China2008 EarthquakeStudy design: Randomised control trial.N = 30. Adults affected by the earthquakeMean age: 53.63 yearsExperimental Group/s: intervention NET *n* = 10, NET-R *n* = 10; waitlist control *n* = 10Narrative Exposure Therapy (NET)Waitlist controlImpact of Event Scale-Revised (IES-R) (Weiss, 2007) [32]7. (Jiang, 2014) [36]China2008 EarthquakeStudy design: Randomised control trial.N = 49.Mean age: 29.8 yearsExperimental Group/s: intervention *n* = 27; treat as usual *n* = 22Interpersonal Psychotherapy (IPT), a 12-week structured psychotherapy TAUClinician-Administered PTSD Scale (CAPS; Blake et al., 1995) [26]8. (Jacob, 2014) [37]Rwanda1994 Rwandan genocideStudy design: Randomised control trial.N = 76.Typical age: 47.55 years (widow); 24.55 years (orphan)Experimental Group/s: intervention *n* = 38; waitlist control *n* = 38Narrative exposure therapy (NET) treatmentWaitlist controlClinician-Administered PTSD Scale (CAPS)9. (Ruggiero, 2015) [38]USATornadoes Study design: Randomised control trialN = 2000. Adolescents and parents from communities affected by tornadoesMean age: 14.55 years (12–17 years)Experimental Group/s: intervention BBN *n* = 364, BBN + ASH *n* = 366; control *n* = 257Bounce Back Now (BBN), a modular web-based intervention;BBN plus a 7-module adult self-help (ASH) interventionNo treatmentAdolescent symptoms of post-traumatic stress disorder (PTSD) and depression assessed using the National Survey of Adolescents (NSA) PTSD module (Kilpatrick et al., 2003) [39]10. (Bass, 2016) [40]IraqConflict and displacementStudy design: Randomised control trialN = 209.Population in the northern Dohuk regionMean age: 40 years (18–82 years)Experimental Group/s: intervention *n* = 159; control *n* = 50A 2-week training program that emphasized a social work model of helping and support.Waitlist controlThe Hopkins Symptom Checklist-25 (HSCL-2528,29 (a 25-item version of the HSCL) for symptoms of depression and anxiety (American Psychiatric Association, 1994) [41]11. (Acarturk, 2016) [11]SyriaHumanitarian traumaStudy design: Randomised control trialN = 70. Adult refugees located in Kilis Refugee Camp at the Turkish–Syrian border with a PTSD diagnosisMean age: 33.7 yearsExperimental Group/s: intervention *n* = 37; control *n* = 30EMDRWaitlist controlIES-R (a self-report instrument which rates the severity of PTSD symptoms) (Weiss, 2007) [32]12. (Cohen, 2017) [42]USAHurricane Study design: Simulation model.N = 2,642,713 (Model population living in the areas of New York City affected by Hurricane Sandy).Typical age: 33.9% aged 18–34 years, 49.0% aged 35–64 years, and 17.1% aged <65 years)Experimental Group/s: n/aStepped care case-finding intervention (stepped care (SC))Usual careThe Child Post-Traumatic Stress Disorder Reaction Index (CPTSD-RI) (Frederick et al., 1992) [43]13. (Dawson, 2018) [44]IndonesiaCivil conflictStudy design: Randomised control trial.N = 64. Children with post-traumatic stress disorder.Mean age: 10.7 years. (7–14 years)Experimental Group/s: intervention *n* = 32; comparison *n* = 32Trauma-focused cognitive behavior therapyProblem-solving therapy provided by lay counselors who were provided with brief trainingPTSD measured by the PTSD Child Reaction Index. (Frederick et al., 1992) [43]14. (Welton-Mitchell, 2018) [45]NepalEarthquakeStudy design: Quasi-experimental study. Cluster randomised design.N = 240. People in two earthquake-affected communitiesMean age: 38 years.Experimental Group/s: intervention *n* = 120; control *n* = 120A hybrid mental health and disaster preparedness interventionWaitlist controlA 7-item investigator developed checklist to measure self-reported disaster preparedness15. (James, 2020) [46]HaitiEarthquake; hurricanes, flooding, and landslidesStudy design: Randomised control trial.N = 480. Community members, drawn from three disaster-affected communitiesMean age: 37 years (18–78).Experimental Group/s: intervention *n* = 240; control *n* = 240Community-based mental health-integrated disaster preparedness interventionWaitlist controlA 12-item Humanitarian Emergency Settings Perceived Needs (HESPER) developed by WHO and King’s College London (Semrau et al., 2012) [47]16. (Rahman, 2019) [38]PakistanArmed conflictStudy design: Cluster randomised control trial.N = 612. Women in a post-conflict setting.Mean age: 36.32 years.Experimental Group/s: intervention *n* = 306; control *n* = 306Brief group psychological interventionEnhanced usual care (EUC)Hospital Anxiety and Depression Scale (HADS) (Zigmond and Snaith, 1983) [48]17. (Dhital, 2019) [49]Nepal EarthquakeStudy design: Cluster randomised control trial.N = 1220. Students from 15 selected schools.Typical age: School-going adolescents from grades six to eight.Experimental Group/s: intervention *n* = 605; control *n* = 615Psycho-social support provided by trained teachersNo treatmentPTSD symptoms were measured using CPSS, a 17-item measure for children and adolescents (Foa et al., 2001) [50]18. (Kılıç and Şimşek, 2019) [51]TurkeyDisasterStudy design: Randomised control trial.N = 76. Nursing students.Typical age: 81.6% 20–23 years; 18.4% 24 years and above.Experimental Group/s: intervention *n* = 38; control *n* = 38.Psychological first-aid trainingNo treatmentDisaster preparedness perception scale (Özcan, 2013) [52]19. (Sijbrandij, 2020) [53]Sierra LeoneEbola and disastersStudy design: Cluster randomised control trial.N = 408. Primary health workers.Mean age: 39.5 years (intervention); 38.5 years (control).Experimental Group/s: intervention *n* = 202; waitlist control *n* = 206.One-day face-to-face PFA group trainingsWaitlist controlPost-PFA assessment; professional attitude20. (Hamdani, 2020) [54]PakistanConflict and displacementStudy design: Randomised control trial.N = 346. Primary care attendees with high levels of psychological distressTypical age: n/a.Experimental Group/s: intervention *n* = 172; comparison *n* = 174Problem managementEnhanced TAUHospital Anxiety and Depression Scale (HADS) (Zigmond and Snaith,1983) [48]Incremental costs per unit change in anxiety, depression, and functioning scores.21. (Lotzin et al., 2021) [55]Germany Traumatic event: life threatening illness or injury, etc.Study design: Randomised control trial.N = 30. Survivors.Mean age: 42 years (22–63 years).Experimental Group/s: intervention *n* = 15; waitlist control *n* = 15SOLAR, a 5-session psychosocial interventionWaitlist controlThe Client Satisfaction Questionnaire (CSQ-8) (CSQ-8; Larsen et al., 1979) [56]Abbreviations: BBN—Bounce Back Now; CAPS—The Clinician-Administered PTSD Scale; CHWs—Community health workers; EMDR—Eye movement desensitization and reprocessing; EUC—The enhanced usual care; HESPER—Humanitarian Emergency Settings Perceived Needs; IPT—Interpersonal Psychotherapy; NET—Narrative Exposure Therapy; PTSD—Post-traumatic stress disorder; RCHC—The Resilience and Coping for the Healthcare Community; RCI—Reaching Children Initiative; SPR—Skills for Psychological Recovery; SOLAR—the Skills for Life Adjustment and Resilience; TAU—Treat as usual; and UC—Usual care.


Meta-analyses using random effects models were conducted in RevMan 5.4 using mean differences and 95% confidence intervals to describe between-group differences for changes in PTSD related symptoms. We also examined the effects of different types of intervention providers on the results by dividing the study participants into three groups: professional-led intervention, non-mental specialist healthcare worker intervention, and internet-based intervention.

### 2.4. Risk of Bias and Quality of Evidence Assessment

Two independent reviewers (MP and JS) assessed the risk of bias using the Cochrane Risk of Bias Tool (ROB2 and ROBIN-I) [57,58]. Any disagreements between the reviewers were resolved through consensus, with reference to the full text. The quality of randomized controlled studies was assessed using the Jadad scale in the following domains: measures, interventions, assignment, confounded conditions, and use of multimodal measures. The domains were confirmed as good, fair, poor, or unclear. A study was assessed as poor quality if it had at least one poor quality domain. If a study was assessed with both unclear and good quality domains, it was assessed as being of unclear quality. A funnel plot was created using RevMan 5.4.1 to show the risk of publication bias.

Quality assessment of the included 21 articles was conducted using the Jadad scale [59] and Newcastle–Ottawa Scale (NOS) [60]. The quality of the randomized controlled studies was assessed in three areas: randomization, blinding, and description of withdrawals/dropouts. The quality of the non-randomized controlled studies was assessed in four areas: selection, comparability, exposure, and outcome. The NOS consists of eight items with three subscales, and the maximum total score for these three subsets is 9. Since there is no consensus on what defines a high-quality study, we considered a study with a score of less than 7 to be of high quality [61].

### 2.5. Outcome Measures

Measures of effect included Perceived Stress Scale (PSS) [62]; self-reported Hospital Anxiety and Depression Scale (HADS) [48]; Depression Self-Rating Scale, which is an 18-item self-report measure for children and adolescents [63]; self-reported behavior changes [64]; Coping Self-Efficacy Scale for Trauma (CSE) [65]; Impact of Event Scale-Revised (IES-R) [1]; Multidimensional Scale of Perceived Social Support (MSPSS) [66]; Clinician Administered PTSD Scale (CAPS) [67]; Structured Clinical Interview for DSM-IV (SCID) [68]; the M.I.N.I. (a semi-structured clinician-rated interview) [69]; and the Post-traumatic Stress Disorder (PTSD) Checklist (PCL) [70].

### 2.6. Data Synthesis

Narrative synthesis was used to report the main characteristics, methods, and findings of the included studies. In addition, a meta-analysis was conducted on studies with primary outcomes centered on PTSD reduction. A mean difference combined with a random-effects model was used to synthesize continuous data. The mean difference, 95% confidence interval, and *p*-value were provided with an associated *I*^2^ measure of heterogeneity. If *I*^2^ > 80%, this result was explored using the following subgroups to explain the heterogeneity: age, frailty, intervention type, sex, study size, and study-level quality assessment.

## 3. Results

### 3.1. Study Characteristics

A total of 151 reports were identified from the electronic database searches after removing duplicate records. After screening titles and abstracts for relevance, 58 full texts were assessed to yield the 21 studies included in the review. The remaining 21 studies were quality-assessed and included in the final synthesis. Of the included studies, 16 were randomized controlled trials, three were controlled studies [27,34,71], and two were cost effectiveness analyses [42,54]. See Table 1 for details.

All included studies were of good quality. See Table 2 and Table 3 for details.
ijerph-19-15847-t002_Table 2Table 2Quality Assessment of randomised controlled studies using the Jadad Scale.Author#1 (0–2)Randomisation#2 (0–2)Masking (Blinding)#3 (0–1)Withdrawals and Dropouts (Accountability of Participants)Quality ScoreJacob et al. (2014) [37]2215Bass et al. (2016) [40]2215Acarturk et al. (2016) [11]2215Basoglu et al. (2005) [14]2204Jiang et al. (2014) [36]2204Dawson et al. (2018) [44]2204James et al. (2020) [46]2204Steinmetz et al. (2012) [29]2114Ruggiero et al. (2015) [38]2204Rahman et al. (2019) [38]2204Sijbrandij et al. (2019) [53]1214Dhital et al. (2019) [49]2114Kilic and Şimşek (2019) [51]2114Hamdani et al. (2020) [54]2114Lotzin et al. (2021) [55]2114Zang et al. (2013) [31]2114Zang et al. (2014) [35]2114
ijerph-19-15847-t003_Table 3Table 3Quality Assessment of non-randomised studies using the Newcastle–Ottawa Scale (NOS) assessment tool [60].Author#1Selection#2Comparability#3Exposure/OutcomeLevel of Quality Welton-Mitchell et al. (2018) [45]323GoodWolmer et al. (2005) [27]322GoodWade et al. (2014) [72]322GoodMcCabe et al. (2011) [73]322GoodPowell and Yuma-Guerrero (2016) [74]322GoodO’Donnell et al. (2020) [4]322GoodAdams et al. (2013) [34]223GoodCohen et al. (2017) [42]422Good(Thresholds: Good quality—3 or 4 stars in selection domain AND 1 or 2 stars in comparability domain AND 2 or 3 stars in outcome/exposure domain; Fair quality—2 stars in selection domain AND 1 or 2 stars in comparability domain AND 2 or 3 stars in outcome/exposure domain; and Poor quality—0 or 1 star in selection domain OR 0 stars in comparability domain OR 0 or 1 star in outcome/exposure domain).


### 3.2. Characteristics of the Included Studies

More than half (9/17) of the 17 RCT studies used a waitlist/delayed-treatment control group. The majority (14/17, 82.4%) of studies were carried out in low- and middle-income countries (LAMICs)—China (3), Turkey (2), Pakistan (2), Haiti (1), Syria (1), Iraq (1), Rwanda (1), Nepal (1), Indonesia (1), and Sierra Leone (1)—while only 17.6% originated from upper-middle-income countries—USA (2) and Germany (1).

Regarding intervention type, six were psychotherapies, including three using narrative exposure therapy (NET) and one each using eye movement desensitization and reprocessing (EMDR), cognitive behavioral technique (CBT), and interpersonal psychotherapy (IPT). The duration of psychological intervention for affected people varied from 1 h behavior treatment [14] to two to three-hour psychotherapy [37] to five weekly sessions of psychosocial intervention [44,55] and 12 h psychotherapy [36]. The length of mental health training intervention for providers (practitioners) ranged from one day [34,53] to 6 h training [51] to one or two weeks [38,44] to a maximum period of 40 h, as, for example, in Problem Management Plus (PM+) [54].

Most studies (19/21) investigated outcomes for disaster survivors covering earthquakes (8), hurricanes (2), tornadoes (1), flooding and landslides (1), conflict and displacement (6), public health emergency (1), and two reported outcomes for community workers, primary health care workers, and emergency response personnel.

In terms of sample size, 14.3% of the studies had small samples (*n* < 30), most of the studies (66.7%) were between 31 and 600, and 19.0% were above 600.

Five (23.8%) long-term follow-up studies with one year or more were found [14,27,37,38] including one simulated long-term follow-up [42]. Six (28.6%) underwent 3–12-month follow-ups [35,36,38,40,44,46]. Three (14.3%) had follow-ups that lasted less than three months [11,29,31].

### 3.3. Overall Findings of RCTs and Controlled Studies

#### 3.3.1. Effectiveness of the Interventions

##### Psychotherapies

We found seven studies that investigated this type of intervention—they found it was largely effective in all studies [11,14,31,35,36,37,44]. Psychotherapy and narrative exposure therapy significantly reduced PTSD symptoms in adult refugees living in the Kilis Refugee Camp at the Turkish–Syrian border [11], as well as in Rwandan widows and orphans intervened by psychologists [37], when compared with the control groups. Interpersonal psychotherapy, administered by mental health professionals, was found to reduce depression in people who continued to suffer mental health impacts from the 2008 Wenchuan earthquake in Sichuan, China [36]. Narrative exposure treatment provided by psychologists had a significant effect on lowering PTSD, depression, and anxiety as well as improving post-traumatic growth and perceived social support in Sichuan earthquake survivors [31,35].

##### Psychoeducation or Trainings

We found eight studies that investigated these types of intervention—they found it was largely effective in eight studies [29,38,38,40,46,51,53,54]. Counselling delivered by community mental health workers in primary health clinics in northern Iraq significantly reduced depression, dysfunction, and anxiety in adults [40]. A brief group psychological intervention administered by non-specialists working with local community health professionals had a significant effect on anxiety, depression, and dysfunction in women aged 18–60 years living in rural villages in Pakistan [38,54]. Psychological first aid training provided by trained mental health nurses significantly improved self-efficacy and perceived preparedness among nurses in Turkey [51], significantly enhanced their knowledge and understanding of appropriate psychosocial responses, and improved skills in primary health care workers in Sierra Leone [53]. When compared with control groups, two web-based psychoeducation interventions had significantly fewer PTSD symptoms [38] and depression [29]. The brief mental-health-integrated disaster preparedness training conducted by trained lay mental health workers in Haiti was effective [46].

##### Psychosocial Support

We found two studies investigating psychosocial support. One found that it was ineffective in reducing PTSD symptoms, depression symptoms, or improving hope among adolescents delivered by trained school instructors [49], while another study found that the control group was more effective in reducing PTSD symptoms compared with the intervention group which provided support by community non-mental health professionals [55].

#### 3.3.2. Cost-Effectiveness of the Interventions

Only two studies [42,54] assessed the cost-effectiveness of psychological skills and behavior technique interventions in disaster settings. A simulation model found that stepped care case-finding with referral to cognitive behavioral therapy for US hurricane survivors was more cost-effective than referral to coping skills alone at an acceptable additional cost per disability-adjusted life year [42]. Problem management interventions using problem-solving and behavioral techniques delivered by trained non-specialist health workers to distressed primary care patients were found to be more expensive but more effective than treatment as usual [54]. They concluded that the intervention was likely to be more cost-effective, although threshold willingness to pay values were problematic in the LAMIC context.

Table 4 presents an overview of the results of the RCTs and controlled studies.

#### 3.3.3. Meta-Analysis

A meta-analysis of eleven PTSD studies was performed. PTSD-focused psychotherapy interventions delivered by professionals were compared with a waitlist control group [11,14,31,35,37] or a usual care control group [36] during the synthesis process. PTSD-focused mental health interventions conducted by non-professionals were compared with waitlist controls [27,46] or a comparison group [44]. PTSD-focused online mental health interventions were compared with no intervention [38] or usual care conditions [29]). The last time-point of data collection was used as the effect size. In those studies that used waitlist control groups, the pre-intervention data of the waitlist group were compared with the post-intervention data of the intervention group. Nine different measures were used to assess PTSD symptoms. See Table 5.

##### Subgroup: Effect of Mental Health Intervention on PTSD-Related Symptoms

Potential publication bias was assessed by visual examination of the funnel plots. See Figure 2.

##### Comparison of Mental-Health-Professional-Delivered Mental Health Interventions versus Control Group

Six studies with 313 participants compared interventions delivered by specialist mental health professionals, including psychotherapists, psychiatrists, and clinical psychologists or social and human service providers who provide support to improve an individual’s mental health or treat mental disorders. The average age of the participants was 23.0 years, and 58.0% were female. A mean difference fitted with a random-effects model was used to synthesize the continuous data. The random effect model was to estimate the impact of a person’s intrinsic and immeasurable qualities. Five studies found that mental healthcare provided by specialists was more successful in reducing PTSD-related symptoms compared with controls. One study found that mental health interventions were ineffective at reducing PTSD [11].

After pooling, we found that interventions delivered by mental health specialists had a clear effect on PTSD reduction, with a standardized MD = 0.99 (95% CI: 0.42–1.57, *I*^2^ = 80, *p* = 0.0007). Thus, compared with the control group, mental health interventions administered by health professionals were beneficial. We fitted professional-delivered intervention subgroups to this finding and were unable to explain the heterogeneity. See Figure 3.

PTSD reduction with professional-delivered mental health interventions compared with PTSD reduction with control group. Effect sizes (Std diff in means) were computed for control designs. 0.20 = small, 0.50 = medium, and 0.80 = large [62].

##### Comparison of Non-Specialists’ Mental Health Interventions versus PTSD Reduction without Mental Health Interventions

Three studies with 831 participants compared these outcomes. After pooling, we found that the SMD was 0.25 (95% CI: 0.11–0.39; *I*^2^ = 0%, *p* = 0.0007). The effectiveness of non-specialist mental health interventions was shown to be small compared with waitlist controls. One of the three included studies found mental-health-integrated disaster preparedness intervention to be more effective in reducing PTSD-related symptoms [46], while the other two found no significant effect of trauma-focused cognitive behavior therapy and a school-based intervention programme which combined psychoeducation and cognitive behavioral techniques in improving the symptoms of post-trauma compared with the control groups [27,44]. See Figure 3.

##### Comparison of PTSD-Related Symptoms with Web Mental Health Interventions versus PTSD-Related Symptoms without Mental Health Interventions

Two studies with 658 participants evaluated web-based interventions. After pooling, the SMD was 0.19 (95% CI: 0.14–0.35; *I*^2^ = 0%, *p* = 0.01), suggesting that mental health interventions delivered via the Internet were only marginally effective. One study found that mental health interventions were more effective in reducing PTSD [38], while another study found that mental health interventions were not effective [29]. The latter had a limited sample size of 37 [29] compared with those that identified an effect, which had a median of 188 participants. Consequently, a lack of significant effect was associated with a lack of power. See Figure 3.

## 4. Discussion

This is the first systematic review to comprehensively explore both effectiveness and cost-effectiveness of the full range of mental health interventions delivered by frontline health care workers in disaster and emergency contexts. We found evidence of a strong and significant association between mental health interventions provided by frontline specialist health care workers and reduced PTSD symptoms in survivors of disasters and health emergencies. We identified a lack of health economic evidence needed to support decision-making and public investment in enhancing mental health skills training in disasters and public health emergencies. These findings support scaling up the timely and effective mental health interventions to enhance mental health capability of responders in disasters and emergencies and better support people in need.

This systematic review identified 21 studies, including 17 randomized controlled trials and four non-randomized studies, evaluating the effectiveness and cost-effectiveness of mental health interventions delivered by frontline healthcare workers. Only two of these studies assessed the cost-effectiveness of the interventions. The vast majority of RCTs (15/17, 88%) found an improvement in mental health outcomes. Most of these studies examined the primary outcome of PTSD reduction (11/17, 65%). Owing to the heterogeneity of the identified studies, a meta-analysis could not be performed. However, it was possible to implement a meta-analysis of 11 studies on interventions targeted at post-disaster traumatic symptoms and PTSD. This meta-analysis found a strong and significant association between mental health interventions provided by frontline specialist health care workers and reduced PTSD symptoms in survivors of disasters and health emergencies but only small or marginal effects for mental health interventions delivered by non-specialists or web-based, respectively.

There is a need for more studies on mental health interventions for various groups with specific requirements, such as young people, women, the disabled, and the elderly [27,38,38,44]. The results of studies focused on school aged adolescents and children were mixed. School-based mental health interventions targeting children and their educational environment significantly improved overall daily functioning but not trauma-related symptoms [27,44]. The latter could be explained by the fact that it used problem-solving as a comparison instead of no intervention [44]. However, trained teachers’ psychosocial support has no significant effect on children’s PTSD or depression symptoms [49], implying that closer collaboration between mental health professionals and teachers, as well as more specific training, is required for school-based interventions to be both feasible and sustainable [27].

Nearly three-quarters of the studies (71.4%, 15/21) focused on low- and middle-income countries (LAMIC), with RCTs accounting for the majority (13/15). They discovered that a simple supportive counselling and psychoeducation group-based program significantly reduced depression [38,40,54], decreased anxiety [38,54], and improved daily functioning [54]. Moreover, low intensity psychoeducational intervention significantly improved disaster-responding knowledge and skills [53] and general self-efficacy [51]. Psychotherapies such as NET, interpersonal IPT, behavioral treatment, and EMDR have been shown to significantly reduce PTSD symptoms [11,14,31,36,37,44] and major depressive disorders [31,36].

Low intensity and easy-to-implement psychological interventions were also more acceptable for survivors following disasters, as well as more likely to be delivered by trained non-mental health frontline workers, and significantly reduced distress, PTSD symptoms, and functional impairment [55]. Such “low-intensity intervention” or support by community-based providers with less mental health expertise has had a big impact on providing accessible mental health support in resource-poor settings in humanitarian circumstances [35,37,38,40,46,71]. Furthermore, because hiring local trainers/supervisors can reduce costs, local mental health capacity-building is recommended. For disaster response and recovery, multi-sector collaboration and coordination of psychological techniques as well as community-based efforts have been advocated [71]. Additional resources should also be assigned to improve preparedness and response capabilities [46,71].

This review included only publications written in English. For the data synthesis, the narrative methodology for whole sampling may duplicate evidence and risk subjective analysis of studies. Despite these limitations, narrative reviews offer a breadth of literature coverage and flexibility to deal with evolving knowledge and concepts.

## 5. Conclusions

This systematic review synthesised randomized controlled trials and cost-effectiveness studies as key evidence for assessing psychological and mental health interventions delivered by frontline healthcare workers in disaster contexts. Mental health interventions delivered by frontline healthcare workers may be time- and cost-effective for lowering psychological distress in natural disaster settings [44,46]. This review demonstrates that brief and low-intensity interventions, such as psychoeducation and social support programs, as well as skills training, are viable options in disasters.

More controlled trials with adequately powered sample sizes and longer-term follow-ups are needed. Given the inherent difficulty in conducting RCTs among disaster survivors in LAMICs, economic modelling may be particularly useful to estimate cost effectiveness at the population level.

## Figures and Tables

**Figure 1 ijerph-19-15847-f001:**
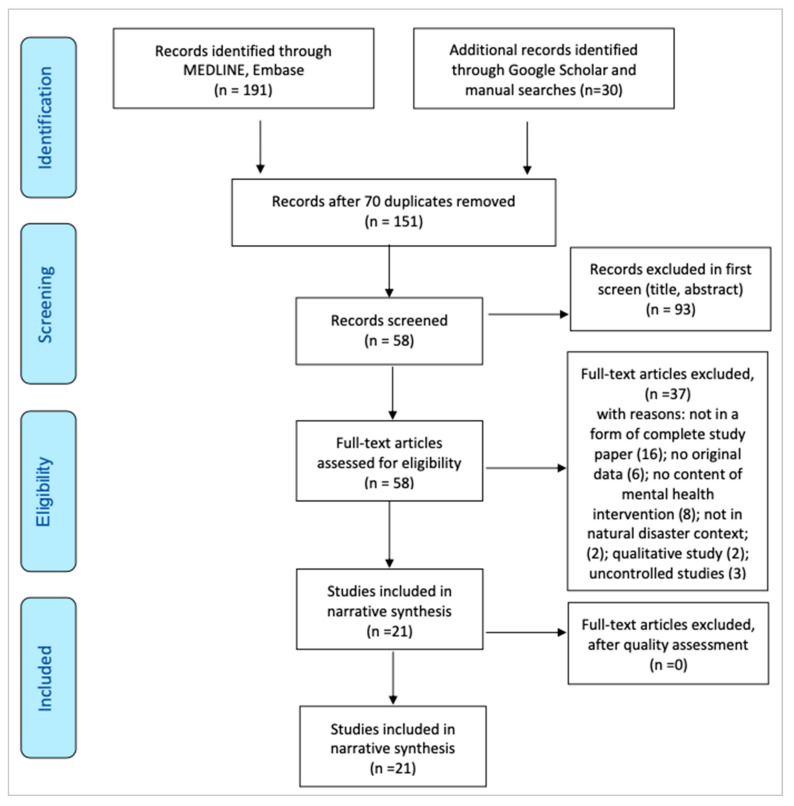
PRISMA Flow Diagram.

**Figure 2 ijerph-19-15847-f002:**
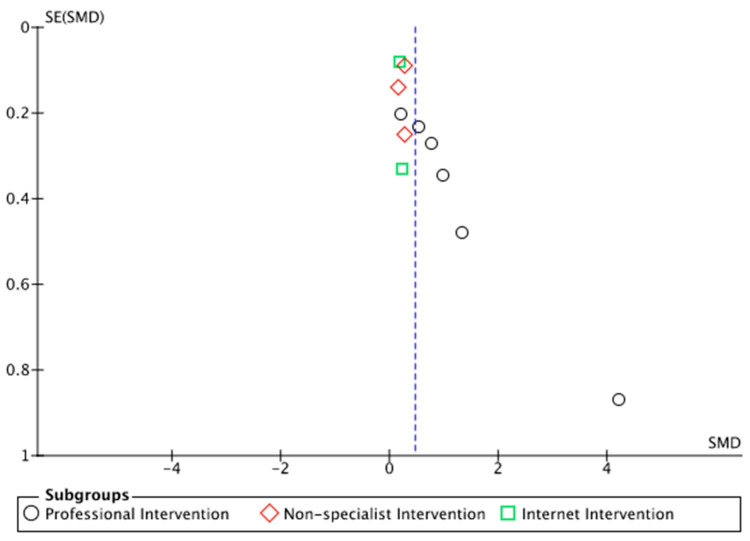
The funnel plot of subgroup of PTSD outcome studies.

**Figure 3 ijerph-19-15847-f003:**
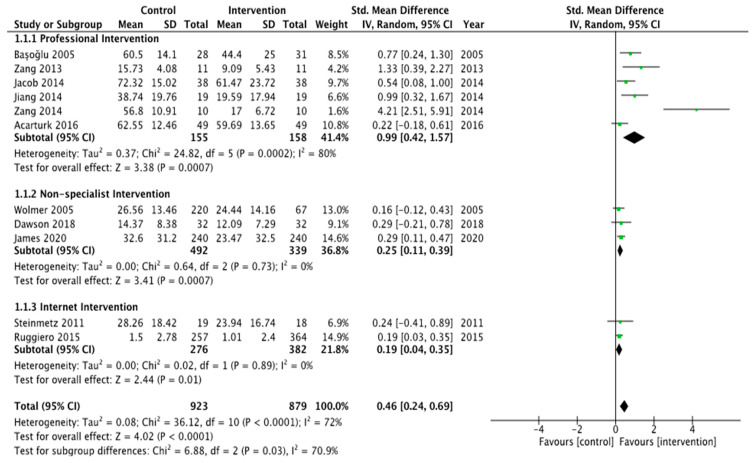
PTSD reduction with professional-delivered mental health interventions compared with PTSD reduction with control group [11,14,27,29,31,35,36,37,38,44,46].

**Table 4 ijerph-19-15847-t004:** Summary of Findings of RCTs.

Author and Year of Study	Findings Reduced Depression	Reduced Anxiety	PTSD Symptoms	Self-Efficacy	Perceived Support	Other
Steinmetz et al. (2012) [29]	++	N/A	N/A	N/A	N/A	Worry ++
Zang et al. (2013) [31]	++	++	++	N/A	N/A	
Zang et al. (2014) [31]	++	++	++	N/A	++	Post-traumatic growth ++
Jacob et al. (2014) [37]	N/A	N/A	++	N/A	N/A	Functional impairment ++
Bass et al. (2016) [40]	++	++	N/A	N/A	N/A	Dysfunction ++
Acarturk et al. (2016) [11]	N/A	N/A	++	N/A	N/A	
Basoglu et al. (2005) [14]	++	N/A	++	N/A	N/A	Fear and avoidance ++
Jiang et al. (2014) [36]	++	N/A	++	N/A	N/A	
Dawson et al. (2018) [44]	+	N/A	++	N/A	N/A	Anger ++
James et al. (2019) [46]	++	++	++	N/A	N/A	Disaster preparedness ++
Ruggiero et al. (2015) [38]	+	N/A	++	N/A	N/A	Alcohol use +
Rahman et al. (2019) [38]	++	++	N/A	N/A	N/A	
Sijbrandij et al. (2019) [53]	N/A	N/A	N/A	N/A	N/A	Knowledge ++
Dhital et al. (2019) [49]	++	N/A	++	N/A	N/A	
Kilic and Şimşek (2019) [51]	N/A	N/A	N/A	++	N/A	Disaster preparedness ++
Hamdani et al. (2020) [54]	++	++	N/A	N/A	N/A	Functional impairment ++
Lotzin et al. (2021) [55]	++	++	N/A	N/A	N/A	Improved coping with the problem +; social support +; quality of life +

++: significantly effective; +: effective but not significant.

**Table 5 ijerph-19-15847-t005:** Summary characteristics of PTSD focused studies, according to study year.

Study (Year)	Participants	Mean Age	Intervention	Comparisons	Follow-up (Months)	Outcome Measure
Başoğlu (2005) [14]	N = 59, 84.7% female; 16–65 years; Turkey earthquake survivors, TSSC score higher than 20, literate.	36.3	SSBT	Waitlist control	24	TSSC
Wolmer (2005) [27]	N = 287, 60.6% female; children aged 9–17 years; students in three schools located in the Turkey earthquake disaster area.	11.5	School reactivation program	No intervention	36	CPTSD-RI
Steinmetz (2011) [29]	N = 56, 85.7% female; Hurricane Ike survivors, had access to the Internet, and met distress criteria	43.0	MDR	Usual care	1	MPSS
Zang (2013) [31]	N = 22, 77.3% female; Adult earthquake survivors seeking assistance, and met the DSM-IV criteria of PTSD	55.73	NET	Waitlist control	2	IES-R
Jacob (2014) [37]	N = 76, 81.82% female children; Rwandan widow and orphan genocide survivors	Widow 47.55; Children 24.55	NET	Waitlist control	6	CAPS
Jiang (2014) [36]	N = 49, 71.4% female; 18 years or older, able to attend weekly sessions, met criteria for PTSD with heavy exposure to earthquake	29.8	IPT	Usual care	6	CAPS
Zang (2014) [35]	N = 30, 93.3% female; earthquake survivor adults met the DSM-IV criteria of PTSD	53.63	NET	Waitlist control	3	IES-R
Ruggiero (2015) [38]	N = 987, 53.5% female; adolescents from communities affected by devastating tornadoes	14.55	BBN	No intervention	12	NSA
Acarturk (2016) [11]	N = 70, 74% female; adult refugees located in Kilis Refugee Camp at the Turkish–Syrian border with a PTSD diagnosis	33.7	EMDR	Waitlist control	1	IES-R
Dawson (2018) [44]	N = 64, 46.9% female; 7–14 years; children living in the region affected by Aceh’s civil conflict and satisfying criteria for probable PTSD	10.7	CBT	Problem-solving intervention	3	UCLA PTSD-RI
James (2019) [46]	N = 480, 49.8% female; 18–78 years; community members, drawn from three disaster-affected communities	37	Mental health integrated disaster preparedness	Waitlist control	6	Unstandardised regression coefficients

## Data Availability

Not applicable.

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
