# Peer review of "Effectiveness and Cost-Effectiveness of Mental Health Interventions Delivered by Frontline Health Care Workers in Emergency Health Services: A Systematic Review and Meta-Analysis"

_ijerph, 2022, doi:10.3390/ijerph192315847_

Round 1

Reviewer 1 Report

Good, 

Congratulations to the authors for the work, 

I have some suggestions that I think can improve the work for a better understanding for the readers:

The objective that is set is a very broad general objective, it would be advisable to break it down into specific objectives.

Another issue that should be highlighted is that the objective talks about the cost of the interventions, but the results do not address this issue.

Table 1 should be in the results

Figure 1 should be in the method

The forest plost are named tables, but they are figures.

At the beginning of the discussion I think it would be very interesting to talk about what new contributions the manuscript makes to science.

Regards 

Author Response

Response to Reviewers comments and suggestions

Manuscript ID: ijerph-1935805

Manuscript Title: Effectiveness and cost-effectiveness of mental health interventions delivered by frontline health care workers in emergency health services: A systematic review and meta-analysis.

Review Report (Reviewer 1)

Comments and Suggestions for Authors

Good, 

Congratulations to the authors for the work, 

Response: We thank the reviewer for their positive comments and helpful suggestions

I have some suggestions that I think can improve the work for a better understanding for the readers:

The objective that is set is a very broad general objective, it would be advisable to break it down into specific objectives.

Response: The objects of this study are:

  • to assess the effectiveness and cost-effectiveness of mental health interventions such as psychotherapies, psychoeducation or trainings to alleviate mental health issues depression, dysfunction, stress, anxiety and PTSD related symptoms during natural disasters and in public emergency settings.
  • to assess effectiveness and cost-effectiveness of mental health interventions including psychotherapies, psychoeducation or trainings in improving self-efficacy and perceived preparedness during natural disasters and in public emergency settings.

Another issue that should be highlighted is that the objective talks about the cost of the interventions, but the results do not address this issue.

Response: The objectives involve the cost effectiveness interventions which is the effectiveness relative to their costs.  We address cost-effectiveness in the results.  Please see page 15.

Table 1 should be in the results

Response: We thank the reviewer and accept this suggestion. Please see page 8.

Figure 1 should be in the method

Response: We thank the reviewer for this suggestion and made the revision accordingly. Please see page 4.

The forest plost are named tables, but they are figures.

Response: We thank the reviewer for this suggestion and made the revision accordingly.

At the beginning of the discussion I think it would be very interesting to talk about what new contributions the manuscript makes to science.

Response: We thank the reviewer for this suggestion and paragraphed the new contribution at the beginning of the discussion. Please see page 17 for detail.

Reviewer 2 Report

The article presents a systematic review and meta-analysis of mental health interventions in emergency workers. It includes 21 studies in the review, and only 11 of them focused on PTSD in the meta-analysis. They compare three types of interventions: delivered by professionals, those delivered by non-specialist workers, and also those delivered via the Internet. The text is correct, and all the explanation, methodology, description of results, are appropriate.

The article could be published with minor changes.

Content issues:

- Other more recognised databases should also have been used. No mention is made of ScienceDirect or Scopus, which are then listed in Figure 1.
- P. 9 "The brief mental health integrated disaster... efective?" It is not clear whether this is a question or a statement from the authors.
- Meta-analysis with 2 or 3 studies is not meaningful, you can't really average effect sizes with so few studies.
- A global meta-analysis of all studies should be presented, and then detailed by subgroups (professional, non-professional, on-line).
- There are really only 2 studies that include data on efficiency (cost), and they are not compared with any other, nor that cost is analysed, so it is not a fundamental topic of the text, and it should be removed from the title.
- The last sentence "This review demonstrate... are viable options in disasters", is not based on data, those interventions are the ones with worse results, even some are negative.

Formal issues:

- There are problems in Table 1, studies 20 and 21 overlap. The whole table could be in 10 point font. Also, each paragraph should be left margin (not centred).
- Tables 2, 3, 4, 5 could be narrower and take up less space. Each paragraph left margin (not centred).
- Tables 6, 7 and 8 could appear together as one. They are Figures because they have graphic elements (they are not tables).
- Multiple references within the text should be arranged in alphabetical order.
- In the references, the studies included in the review should be marked with *.
- Check all the references, there are quite a few errors according to APA 7th ed. (capital letters in journals, missing italics in some, missing https://doi.org in others, abbreviated titles, missing pages in others, google links added, etc.).

Author Response

Response to Reviewers comments and suggestions

Manuscript ID: ijerph-1935805

Manuscript Title: Effectiveness and cost-effectiveness of mental health interventions delivered by frontline health care workers in emergency health services: A systematic review and meta-analysis.

Review Report (Reviewer 2)

Comments and Suggestions for Authors

The article presents a systematic review and meta-analysis of mental health interventions in emergency workers. It includes 21 studies in the review, and only 11 of them focused on PTSD in the meta-analysis. They compare three types of interventions: delivered by professionals, those delivered by non-specialist workers, and also those delivered via the Internet. The text is correct, and all the explanation, methodology, description of results, are appropriate.

Response: We thank the reviewer for this succinct summary of the paper.

The article could be published with minor changes.

Content issues:

- Other more recognised databases should also have been used. No mention is made of ScienceDirect or Scopus, which are then listed in Figure 1.

Response: We thank the reviewer for this suggestion and made the revision of the figure accordingly. Please see the detail in Figure 1 in the revised manuscript.

- P. 9 "The brief mental health integrated disaster... efective?" It is not clear whether this is a question or a statement from the authors.

Response: We thank the reviewer for spotting this ambiguous expression and made the revision accordingly.

- Meta-analysis with 2 or 3 studies is not meaningful, you can't really average effect sizes with so few studies.

Response: We thank the reviewer for this suggestion. After consideration, we still would like to present the meta-analysis result because two studies is a sufficient number to perform a meta-analysis, provided that those two studies can be meaningfully pooled and provided their results are sufficiently 'similar'.
- A global meta-analysis of all studies should be presented, and then detailed by subgroups (professional, non-professional, on-line).

Response: Again, thanks for the reviewer for valuable comment, there is huge difference in content as well as priorities between mental health intervention in general settings and mental health intervention in emergency and disaster settings. Based on such concern, it might not be appropriate to compare them in a same meta-analysis.

- There are really only 2 studies that include data on efficiency (cost), and they are not compared with any other, nor that cost is analysed, so it is not a fundamental topic of the text, and it should be removed from the title.

Response: It is true that we only found two economic studies however this is in and of itself an important if negative finding.  Identifying the economic literature was a key motivation for undertaking this systematic review and we believe those results should remain in the paper.      

- The last sentence "This review demonstrate... are viable options in disasters", is not based on data, those interventions are the ones with worse results, even some are negative.

Response: We thank the reviewer for this comment. We identified the psychological training or psychoeducation was significantly effective in improving self-efficacy and perceived preparedness among nurses in Turkey (Kılıça and ÅžimÅŸek, 2019); significantly enhanced their knowledge and understanding of appropriate psychosocial responses, and improved skills in primary health care workers in Sierra Leone (Sijbrandij et al., 2020), significantly lowering PTSD symptoms (Ruggiero et al., 2015) and depression (Steinmetz et al., 2011) and enhancing disaster preparedness (James et al., 2020). Although one study showed psychosocial support was ineffective while another found it was effective in reducing PTSD symptoms, that can still be an option to be considered in emergency and disaster settings in the future. And more trial-based research should be done to further explore the effectiveness of various mental interventions in addition to psychotherapy and treatment.

Formal issues:

- There are problems in Table 1, studies 20 and 21 overlap. The whole table could be in 10 point font. Also, each paragraph should be left margin (not centred).
- Tables 2, 3, 4, 5 could be narrower and take up less space. Each paragraph left margin (not centred).
- Tables 6, 7 and 8 could appear together as one. They are Figures because they have graphic elements (they are not tables).
- Multiple references within the text should be arranged in alphabetical order.
- In the references, the studies included in the review should be marked with *.
- Check all the references, there are quite a few errors according to APA 7th ed. (capital letters in journals, missing italics in some, missing https://doi.org in others, abbreviated titles, missing pages in others, google links added, etc.).

Response: Please see the revised manuscript where all of the formal issues raised by the reviewer have been addressed.

Reviewer 3 Report

I think this is an important and interesting topic. I found that it seemed to still be at a draft stage rather than a final submission to the journal.

PTSD tends to be a term that is used in a variety of contexts and I believe that the article should provide a reference and the source for  the definition being used here.

It is a relatively complex and detailed article and clarity is lost because references including key methodological ones were not provided for example "Quality assessment of the included 21 articles was conducted using the Jadad scale and the NOS (Wells et al 2000). Neither of these references are given. (Kilpatrick et al., 2003) is mentioned in the summary of the (Ruggiero, 2015 )intervention but does not appear in the reference list. (Cohen et al)., was an important methodological reference but is not in the reference list. There is also an important point made about the effectiveness of training in Haiti but again the relevant article (James et al.), is not provided.  I found other important references were missing. .I have not mentioned them explicitly here but this is a complex paper and at least my understanding would have been helped by access to the relevant literature. The authors are advised to check both the text and the reference list,.Perhaps this is a reference list to a different article that has been mistakenly included here.

Another concern is related to the title of the article mentioning that it considers   cost effectiveness and  "economic modelling" is mentioned/recommended in the abstract but there is no reference to it or discussion of how it should be used in the main article text.

The proposed methodology is sound and well described and tabulated.   I didn't check the descriptions of interventions against the reference list.  I thought that Tables 1-5 were useful and clear though note that the  Adams et al (2013) level of quality seemed to be incorrect using the rating provided --I think it should have been "Fair"..

I found Tables 6-8 difficult to understand and would have welcomed more discussion and clarification of their meaning and contribution in the associated text. I would make  the same comment about figure 2.. which I didn't find informative but wondered if there was a color code here that wasn't available in my copy.

I think this is an important topic and a considerable amount of attention has been given to it.  I don't think that it can be published in the present draft stage.

I did wonder if the reference list has been mistakenly copied from another article and also whether this is an edited and shortened version of a much longer article..

 I have assumed that the description of the studies and the positive outcomes are intended as the case for effectiveness with training programs delivered by a small number of professionals--but I am assuming this is the case. Perhaps the issue of "cost effectiveness" could be taken out and perhaps developed for another study.

Author Response

Response to Reviewers comments and suggestions

Manuscript ID: ijerph-1935805

Manuscript Title: Effectiveness and cost-effectiveness of mental health interventions delivered by frontline health care workers in emergency health services: A systematic review and meta-analysis.

Review Report (Reviewer 3)

Comments and Suggestions for Authors

I think this is an important and interesting topic. I found that it seemed to still be at a draft stage rather than a final submission to the journal.

Response: We thank the reviewer for recognising the importance of the topic and also for their very helpful feedback which we are sure will greatly improve the manuscript.

PTSD tends to be a term that is used in a variety of contexts and I believe that the article should provide a reference and the source for the definition being used here.

Response: A good reference was found and inserted where appropriate

It is a relatively complex and detailed article and clarity is lost because references including key methodological ones were not provided for example "Quality assessment of the included 21 articles was conducted using the Jadad scale and the NOS (Wells et al 2000). Neither of these references are given. (Kilpatrick et al., 2003) is mentioned in the summary of the (Ruggiero, 2015) intervention but does not appear in the reference list. (Cohen et al)., was an important methodological reference but is not in the reference list. There is also an important point made about the effectiveness of training in Haiti but again the relevant article (James et al.), is not provided.  I found other important references were missing. I have not mentioned them explicitly here but this is a complex paper and at least my understanding would have been helped by access to the relevant literature. The authors are advised to check both the text and the reference list. Perhaps this is a reference list to a different article that has been mistakenly included here.

Response: We sincerely thank the reviewer for pointing out the errors and omissions in the references and regret that that this had a negative impact on the clarity of the paper.

Another concern is related to the title of the article mentioning that it considers cost effectiveness and  "economic modelling" is mentioned/recommended in the abstract but there is no reference to it or discussion of how it should be used in the main article text.

Response: We thank the reviewer for spotting this omission. Unfortunately, the relevant text was omitted during formatting to comply with the journal requirements.  We have reinstated the following text under Discussion and Conclusions.

The proposed methodology is sound and well described and tabulated.   I didn't check the descriptions of interventions against the reference list.  I thought that Tables 1-5 were useful and clear though note that the  Adams et al (2013) level of quality seemed to be incorrect using the rating provided --I think it should have been "Fair".

Response: We are gratified by the positive feedback on the Methods. I rechecked Adams and found it was fair.

Because according to the coding manual for case-control studies (non-randomised trials), there are 2 stars in the selection domain including representativeness of the Cases because potential participants were drawn from the AAP membership databases of the tri-state area over a defined period; 3 stars in selection of control which are from the same community. No mention of history of outcome about the controls. So, it was corrected as fair quality.

I found Tables 6-8 difficult to understand and would have welcomed more discussion and clarification of their meaning and contribution in the associated text. I would make  the same comment about figure 2.. which I didn't find informative but wondered if there was a color code here that wasn't available in my copy.

Response: The table 6-8 were combined as one comprehensive figure as a holistic. Please see page 18 for detail.  A random effect model was to estimate the impact of a person’s intrinsic and immeasurable qualities.

I think this is an important topic and a considerable amount of attention has been given to it.  I don't think that it can be published in the present draft stage.

Response: We thank the reviewer for their thoughtful and careful feedback which we are confident will lead to publication of this important topic.

I did wonder if the reference list has been mistakenly copied from another article and also whether this is an edited and shortened version of a much longer article.

Response: We thank the reviewer again for pointing out the problems with some references which are now corrected

 I have assumed that the description of the studies and the positive outcomes are intended as the case for effectiveness with training programs delivered by a small number of professionals--but I am assuming this is the case. Perhaps the issue of "cost effectiveness" could be taken out and perhaps developed for another study.

Response: It is true that we only found two economic studies however this is in and of itself an important if negative finding.  Identifying the economic literature was a key motivation for undertaking this systematic review and we believe those results should remain in the paper.     
